

# Ice crystals images from Optical Array Probes: classification with Convolutional Neural Networks

Louis Jaffeux[1], Alfons Schwarzenböck[1], Pierre Coutris[1], and Christophe Duroure[1]

[1]Laboratoire de Météorologie Physique (UMR6016) / UCA / CNRS, Aubière, France

**Correspondence:** Louis Jaffeux (Louis.Jaffeux@uca.fr)

**Abstract.** Although airborne optical array probes (OAP) have existed for decades, our ability to maximize extraction of meaningful morphological information out of the images produced by these probes has been limited by the lack of automatic, unbiased and reliable classification tools. The present study describes a methodology for automatic ice crystal recognition using innovative machine learning. Convolutional Neural Network (CNN) have recently been perfected for computer vision and have been chosen as the method to achieve the best results together with the use of finely tuned dropout layers. For the purposes of this study, CNN has been adapted for the Precipitation Imaging Probe (PIP) and the 2DS-Stereo Probe (2DS), two commonly used probes that differ in pixel resolution and measurable maximum size range for hydrometeors. Six morphological crystal classes have been defined for the PIP and eight crystal classes and an artifact class, for the 2DS. The PIP and 2DS classifications have five common classes. In total more than 8000 images from both instruments have been manually labelled, thus allowing for the initial training. For each probe the classification design tries to account for the three primary ice crystal growth processes: vapor deposition, riming and aggregation. We included classes such as fragile aggregates and rimed aggregates with high intra-class shape variability and commonly found in convective clouds. The trained network is finally tested through human random inspections on actual data to show its real performance in comparison to what humans can achieve.

## 1 Introduction

Accurately representing ice clouds in radiative transfer models is extremely challenging due to the high diversity of the crystal habits present in these clouds (Yi et al., 2016). Thus, improving the general understanding of ice cloud feedback in the climate system requires a better understanding of the processes occurring in these clouds (Wyser, 1999). In addition, the impact of atmospheric conditions on microphysical processes and resulting crystal morphologies cannot be studied without having reliable measurements of crystal habits inside ice clouds.

The qualitative observation of ice crystals in clouds in the $20^{th}$ century has led to numerous attempts of their classification into multiple crystal habit categories. For example, Nakaya (1954), Magono and Lee (1966) and more recently Kikuchi et al. (2013) have produced general classifications for natural ice and snow crystals, the latter including 130 sub classes, reflecting the high diversity in shapes one can expect from ice crystals. Related to the classification methodology, scientists have identified three primary pathways of ice crystal growth, namely vapor deposition, riming, and aggregation (Pruppacher and Klett (2010)). The respective role of each of the three processes in the formation of different types of ice crystals has been frequently



addressed, for example for vapor deposition (Bailey and Hallett, 2009), for graupel (Sukovich et al., 2009), and for aggregation (Hobbs et al., 1974). However, since accurate and reliable in situ measurements of natural ice crystal morphology has been very challenging in ice clouds, the processes associated with the formation and evolution of atmospheric ice are still poorly understood (Baumgardner et al., 2012).


Optical Array Probes (OAP) are high frequency airborne imagers commonly used for in-situ observation of ice crystals in clouds. They produce large amounts of ice crystal images with counting statistics that allow to establish particle size distributions within seconds.

Since OAPs were developed in the 1970s (Knollenberg, 1970), several attempts tried to produce high performance classifica-

tion algorithms based on morphological descriptors. While mathematically simple, the feature extraction for pattern recognition of 2D hydrometeor images developed by Rahman et al. (1981) and Duroure (1982) give an insight on how morphological image analysis is useful to automatically categorize OAP images into different classes. Their approach works well with synthetic images of singular crystals that exhibit completely unambiguous orientations and idealized shapes (see Rahman et al. (1981)). In practice, the overwhelming majority of observed ice crystals are not perfectly oriented, undergo multiple microphysical

processes at different levels, including aggregation, and show natural irregularities. Such methods are also limited by the pixel rendering of edges from the probes, which diminishes their performance. These limits were identified and reported in Korolev and Sussman (2000) where a feature-based classification technique was applied to 2DC data. More recently, this technique has been applied to images from the Cloud Particle Imager (CPI), a CCD-based imager with finer resolution and greyscale levels (Lawson et al., 2006; Lindqvist et al., 2012; Woods et al., 2018) based on criteria for 2D pattern recognition. Finally, Praz

et al. (2018) used features from these previous studies and from Praz et al. (2017) in a new methodology called Multinomial-Linear-Regression (MLR) to classify images from two different OAPs (2DS and HVPS) and the CPI. This classification tool has brought the feature-based approach to its highest maturity, but is still very limited in its ability to quickly process and classify images, and furthermore was only roughly evaluated on two 1 min flight periods of the OLYMPEX campaign (Houze et al., 2017). In conclusion, the feature based approach in its ultimate form is not only slow and trained specifically for a given

context, but it operates in a very distant manner to the way our brain identifies shapes and objects, potentially creating bias from feature definitions.

Considering the fact that computer vision has advanced in a way that today it can emulate the human brain's ability to recognize shapes and objects (Russakovsky et al., 2015), a different approach to the classification problem was favored in the present study. Instead of relying on designed features, a widespread and wellknownmethod called Convolutionnal Neural

Network (CNN) (Krizhevsky et al., 2012; He et al., 2016) reproduces the human ability to identify complex shapes and objects and develops hierarchical sets of features from raw labelled data. During the time when the presented work was under development, CNN classification tools emerged for the CPI (Xiao et al., 2019; Przybylo et al., 2021), however they still need to be adapted for OAP image data. In general, OAP images lack textural information (legacy data sets comprises black and white images while newer probes have a maximum of four levels of grey) and also exhibit much coarser resolution (64/128

photodiodes array compared to a 1 mega pixel camera), but have the advantage of continuously imaging the sample volume



between the probes' arms which is not the case when relying on a particle detector with comparably few CCD high resolution greyscale images. As a result, the use of OAP instruments in airborne campaigns produce a more quantitative and statistically meaningful representation of the cloud microphysical state, however with diminished morphological information (3D object projected to binary 2D image) .


In section 2 of this study, the OAP data and chosen morphological classes are presented. Then the CNN methodology for the automatic classification of ice crystals for the 2DS and the PIP probes is detailed in section 3, together with the description of the training process and evaluations of the fully trained networks on the test set. Section 4 presents an evaluation of the performance of the two classification tools with random visual inspections. The conclusions are summarized in section 5.

**2    Data Description (Training data)**

The very first step of the Convolutionnal Neural Network methodology is to build a database, where images are associated with labels by an operator. This procedure implies that classes have to be defined beforehand. In the context of defining morphological classes, three items are mentioned and shortly discussed here:

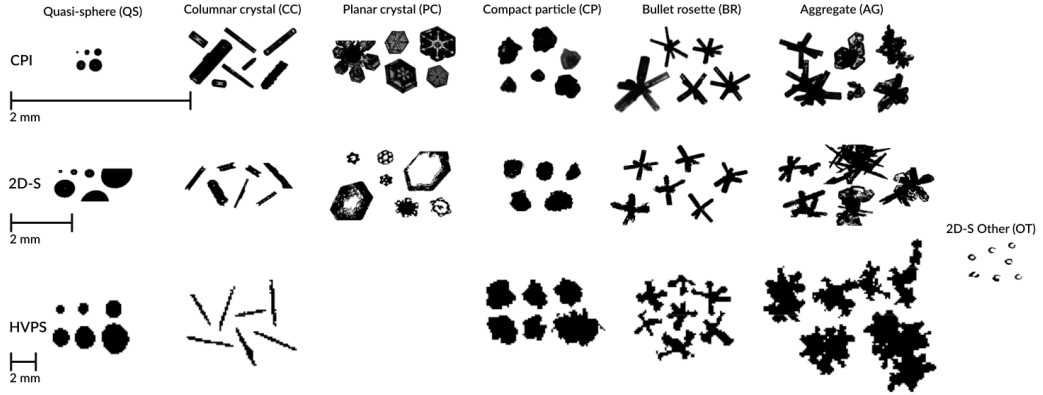

**Figure 1.** Illustration of ice crystals habits from Praz et al. (2018) for different cloud imaging probes

1. The primary goal of our habit classification is to reveal ice crystal growth mechanisms inside a cloud. The designed
classes in this study are rather comparable to those used in Praz et al. (2018) (shown in figure 1) which were themselves inspired by the pioneering work of Magono and Lee (1966). The chosen morphological classes primarily account for the three ice growth mechanisms of vapor deposition, aggregation, and riming. All possible crystal shapes are included in a rather limited number of classes without trying to implement the 130 classes (basically high resolution grey scale CCD images) from Kikuchi et al. (2013).





2. The two probes' technical details are presented in table 1. For the image analysis, only non-truncated images with maximum dimension $D_{max} > 300\mu m$ (30 pixels, for 2DS) and $D_{max} > 2mm$ (20 pixels for PIP) have been classified. Below $300\mu m$, 2DS images are frequently distorted by diffraction effects (e.g Vaillant de Guélis et al. (2019)). This effect persists above this threshold to a lesser extent, and led to the definition of a dedicated artefact class for the 2DS, labeled as Diffracted particles (**Dif**). Heavily rimed aggregates are rather large and thus rarely observed in 2DS images, since they are most likely truncated and thus automatically discarded. Moreover, looking at the 2DS images, strikingly well detailed combinations of columns, plates, and dendrites were found. Although, sometimes it is not clear whether aggregation may have occurred during their formation, the absence of riming and the influence of diffusional growth are undeniable. The corresponding class for those images is denoted Complex Assemblages of plates, columns or dendrites (Complex Assemblages **CA**). The coarse resolution of the PIP makes it practically impossible to discern details such as transparency and sharp edges associated with the diffusional growth. For this reason mixed combinations of columns, plates and dendrites (**CA**) cannot be clearly distinguished from what is designated as fragile aggregates (**FA**). Due to the lower threshold of utilized PIP images of 2 mm, capped columns and water drops are scarce in our training database and thus, were not considered as morphological classes for PIP images in this study.

3. The data used were observed during several airborne research campaigns. Initially HAIC (Dezitter et al., 2013) and EX-AEDRE (Defer et al., 2015) were the main data sources for OAP images. Selecting data and labelling images manually, although being mandatory for a supervised classification scheme, is a long and strenuous process. Some classes were harder to find in these campaigns' data and motivated the use of two further campaigns (AFLUX and EUREC4A) to speed up filling these fewer populated habit classes (see table 2).

**Table 1.** Optical Array Probes technical specifications

| Specifications | 2DS (SPEC Inc.) | PIP (DMT Inc.) |
|---|---|---|
| Frequency | Depends on aircraft speed | Depends on aircraft speed |
| Resolution | $10\mu m$/pixel | $0.1mm$/pixel |
| Number of Photodiodes | 128 | 64 |
| Particle size range | $10\mu m - 1280\mu m$ | $0.1mm - 6.4mm$ |
| Image type | Black and White | Black and White |
| Selected range for classification | $300\mu m - 1280\mu m$ | $2mm - 6.4mm$ |

Traditionally, the training set is comprised of randomly chosen images from the whole available database. Since all the classes are not represented equally in crystal numbers, an adjustment in the loss function should be made to account for the classes with lower representation. Still, an operator in charge of classifying these images would face the difficulty to classify particles from images that stand between multiple classes or that are not identifiable because of ambiguous random projections. Defining a class dedicated to irregular crystals has been avoided, since we believe that, with the high variability associated with crystal shapes, it would be very dependent on the appreciation of the operators who could eventually fit too many images into



**Table 2.** Campaigns used to gather training database

|  | EXAEDRE | HAIC | AFLUX | EUREC4A |
|---|---|---|---|---|
| Reference | Defer et al. (2015) | Dezitter et al. (2013) | not available yet | Bony et al. (2022) |
| OAPs deployed | 2DS and PIP | 2DS and PIP | 2DS | 2DS |
| Crystal habits found | All | All | Col, CBC, HPC | WD |

this "irregular" class. Moreover, the nature of the output of a CNN makes it possible to produce non categorical results, in order to express some level of ambiguity between two or more classes instead of simply stating its inability to identify the image.

The overview of the nine microphysical habit classes accounted for in this study is presented in table 3 and discussed below. Overall, nine morphological classes have been defined, five are common to the two probes: Compact/Graupels (**CP**), Fragile Aggregates (**FA**), Columns (**Co**), Combination of columns and Bullets (**CBC**) and Hexagonal Planar Crystals (**HPC**).

Moreover, one class specific to the PIP consists of Rimed aggregates (**RA**) and three specific classes are added for the 2DS, namely Water Droplets (**WD**), Capped Columns (**CC**), and an artifact class (**Dif**) for out of focus images. **Co**, **CC** and **HPC** are singular, unrimed crystal images that originated solely from diffusional growth. **CBC** and **CA** have mostly grown by deposition of water vapor and may result from aggregation of more than one particle but remain unrimed. **FA** are products of aggregation of several unrimed or lightly rimed particles, while **RA** show an evident fluffy aspect, characteristic of the collection and

freezing of supercooled droplets on the crystal's surface. Finally, **CP** are ice particles with the highest degree of riming, in which the contribution of the two other processes is invisible. In every case, growth by vapor diffusion cannot be ruled out as it continuously contributes to ice production in a cloud.

Some images obtained with OAPs are commonly found to be ambiguous in the sense that they do not clearly belong to exactly one class. One could justify the inability of non-ambiguous classification of every image with two independent ex-

planations. First of all, OAPs are 2D binary low resolution imagers. Random orientations combined with the lack of surface information and the low number of pixels occasionally hide important features that are required to identify certain crystal types. For example, a plate seen from the side could be strictly impossible to differentiate from a column. Secondly, the definition of crystal habit classes is lacunar by design and it is unavoidable that some crystals might be found not to belong to any class or to belong to more than one class. As a matter of fact, the classes defined here or in general in the literature (Kikuchi et al.

(2013) or Magono and Lee (1966)) are only landmarks, local clusters in a continuous multivariate space where ice crystals happen to be moved by the microphysical processes that are active in their respective environment during their lifetime. Taking into account these two factors, it was decided, in the process of forming the initial labelled data set for each probe, that only unambiguous images were selected for the test, validation and training sets, rather than randomly selected images from the available data, trying to classify all of them. Since the classification is meant to be applied on actual data, it is important that

we provide a way to quantify its performance and the uncertainty associated with it (discussed in section 4).





| Class name | Supported probe | Examples 2DS(>300μm) | Examples PIP(>2000μm) | Description | Dominant micro-physical processes | | |
|---|---|---|---|---|---|---|---|
| | | 2mm | 2mm | | Vapor diffusion | Riming | Aggregation |
| Compact particles (CP) | 2DS, PIP | (1053) | (551) | Heavily rimed, compact, Graupels | ● | ＋ | ● |
| Fragile aggregates (FA) | 2DS, PIP | (594) | (627) | Irregular, likely aggregated, unrimed, weak bridges | ● | ● | ＋ |
| Columns and Needles (Co) | 2DS, PIP | (604) | (508) | Singular columns, needles or sheethes | ＋ | － | － |
| Hexagonal planar crystals (HPC) | 2DS, PIP | (414) | (384) | Singular stellar dendrites or plates | ＋ | － | － |
| Rimed Aggregates (RA) | PIP | Truncated on 2DS | (824) | Large, likely aggregated heavily rimed | ● | ＋ | ＋ |
| Combination of bullets or columns (CBC) | 2DS,PIP | (368) | (387) | Bullet rosettes, aggregation or combination of bullet rosette and/or columns. Only aggregated columns and/or rosettes for the PIP | ＋ | － | ● |
| Complex assemblages of planes, columns, dendrites* (CA) | 2DS | (442) | Resolution is not adapted to see small details or render clear edges | Unrimed, aggregated or not, combinations of plates, columns, dendrites | ＋ | － | ● |
| Capped columns (CC) | 2DS | (675) | No capped columns of size >2mm | | ＋ | － | － |
| Water droplets (WD) | 2DS | (482) | No 2mm droplets in ice clouds | | ＋ | ● | ＋ |

**Table 3.** The nine microphysical classes used in the classifications. Green circles mean the micro-physical process recently played a role in the particle's growth. Red circles mean the micro-physical process certainly did not occur in the particle's growth. Gray circle means the micro-physical process might have happened at some point but there is no evidence of it happening recently. In parenthesis, number of images used in the original labelled database for each classes.* *Images shown for combinations of columns, plates and dendrites are scaled down compared to other images so that they fit in the table properly.*





## 3    CNN methodology

This section presents the classification methodology that was applied to the two OAPs. First some insight is given on the implemented Convolutionnal Neural Networks (CNN) technique. Then the training methodology is detailed. Finally, the quality of the training is evaluated on independent test sets and the results are discussed.

### 3.1    Convolutionnal Neural Network, General Principles

CNN and similar Deep Learning techniques are largely used in medical image analysis (Tajbakhsh et al., 2016; Gao et al., 2019), but are also emerging in other research fields, for example in biology for plankton image analysis in Luo et al. (2018). Especially in medical image analysis, the success of CNN algorithms is evident. They are highly reliable and have, by design, the ability to learn hierarchically built complex features from raw data. CNN is therefore an incredibly pertinent technique for image analysis in general Krizhevsky et al. (2012). The following architecture description presents the algorithm in its working state (for further information see Goodfellow et al. (2016)) and its training for each of the two probes will be described in the next subsection. When applied to computer vision, CNN algorithms consist of two parts: a feature extractor and a classifier (see Figure 2a). Both of these have large sets of trainable parameters which will be updated during the learning phase through gradient backward propagation. The feature extractor is hierarchically built with two initial building blocks: convolutional layers (Convlayers) and sub sampling layers (maxPooling in our case), both are illustrated in Figures 2b and 2c, respectively. Convlayers can be seen as filters or masks. In practice, it is a square matrix with trainable values. The size of these filters is called their receptive field (here 3 by 3), and they are applied through a dot product on each pixel and all the pixels around in the receptive field. After normalization and use of an activation function, the convolution of the input by each filter produces a set of feature maps. They are then subsampled with a 2 by 2 max pooling filter. Subsampling diminishes the noise induced by the previous convolution and summarizes the information contained on feature maps to its most crucial part. The output obtained from the subsampling layer is a set of square matrices of dimension twice as small as the input. The number of filters of the next convolution step can therefore be doubled with no increase in computational cost, increasing the potential complexity of the algorithm and, ultimately its ability to generalize and infer relevant abstracted features as we go into the deeper layers. Convlayers and maxpooling layers are repeated (see figure 2a) in the feature extracting part until every feature map is reduced to a 1 by 1 size.

Finally, a fully connected perceptron with one hidden layer serves as the classifier (right side in figure 2a) to attribute a class to the highly abstracted features extracted from the original input image. In this final stage, for individual images probabilities are calculated to belong to any of the eligible classes. A minimum threshold (usually of 50%) can eventually be applied to segregate images that failed to be identified by the algorithm. Actual output model plots for both probes PIP and 2DS are provided in figures A2b and A2a, respectively.

Three state-of-the-art overfitting countermeasures were implemented in the initial architecture: Namely, dropout layers were added in between the subsampling and convolutional layers, an early stopping condition was set during the training phase and batch normalization was applied.





**Figure 2.** CNN Architecture and building blocks

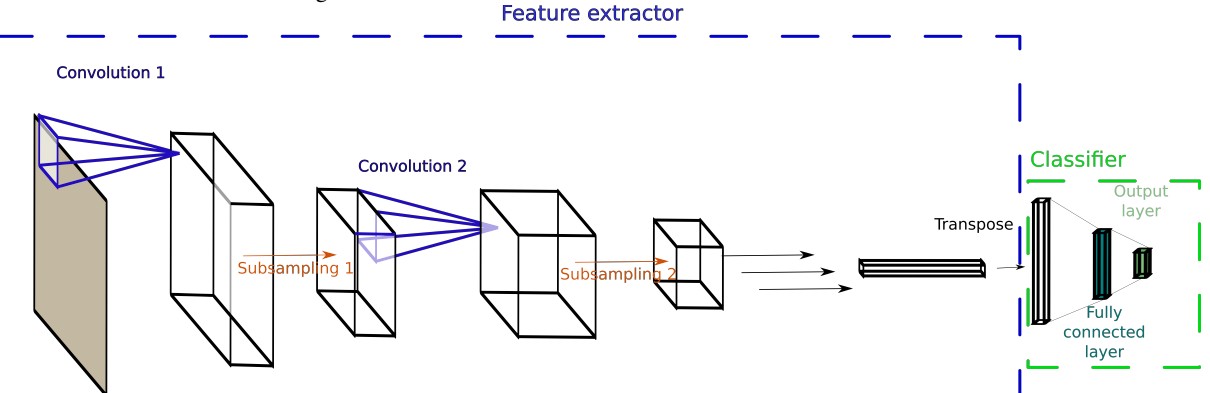

(a) Simplified architecture of Convolutionnal Neural Networks

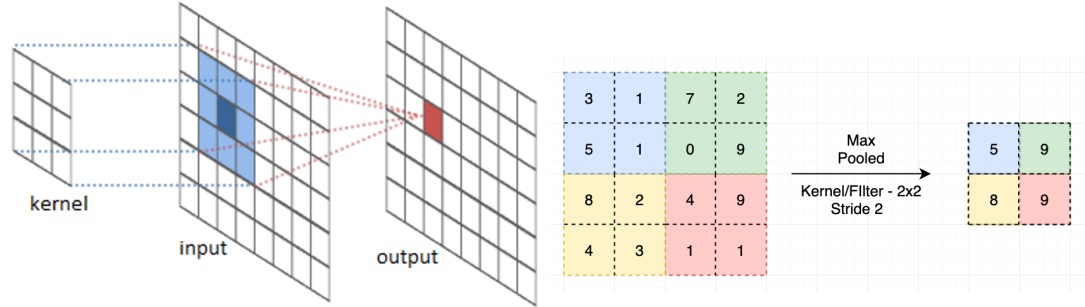

(b) Elementary operation at the heart of convolutions, from the River Trail Documentation

(c) Illustrating maxpooling, found on: medium.com

The use of dropout allows to train very complex models with a limited number of training data without overfitting (Srivastava
et al., 2014). This method is applied only during training on the fully connected layer and on the convolutionnal layers (Park
and Kwak (2016) proved dropout's usefulness on convolutions). An exponential number of shallow models sharing weights
are improved during training. As a result multiple confirmation paths emerge, each one of them focusing on essential features.
The trained model becomes much more robust to noise and translations. The effect of dropout adds to the data augmentation
layer, an early stopping condition and the batch normalization to ensure that overfitting will not happen and that the ability of
the model to generalize is enhanced as much as possible.

## 3.2 Training

An overview of the training methodology is given in figure 3. After labelling the data, the images are padded to the same size
and randomly split into three subsets: test (20%), validation (16%), and training (64%). The training set is used to teach the
model. The feature extractor and the classifier presented in the previous subsection can be trained at the same time using the



feed forward - backward error propagation scheme as represented in figure A1. Every epoch, the model is evaluated on the validation set to monitor its improvements and whether or not overfitting is occurring. Whenever the loss function computed on the validation set fails to improve five epochs in a row, the training is stopped. If the validation loss and accuracy are judged to be satisfactory, we proceed to evaluate the model on the test set. This last step produces performance metrics shown in figure 4 (precision is the fraction of detections reported by the model that were correct, recall is the fraction of true events that were detected, f1-score is the mean of precision and recall (Goodfellow et al., 2016)).

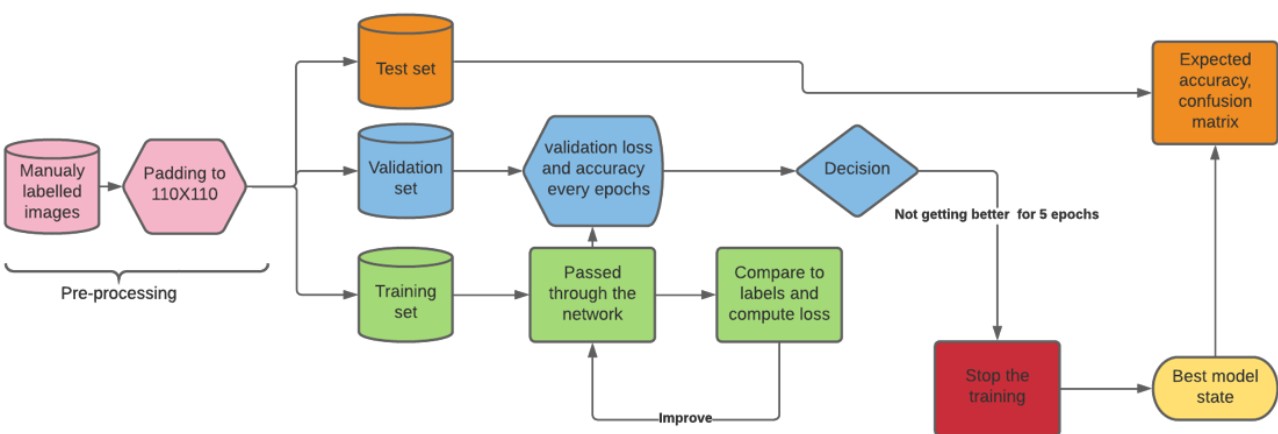

**Figure 3.** Overview of training methodology


### 3.3 Training evaluation: results on test sets

Hyper-parameter tuning was performed using keras inbuilt random search functionnality (Chollet et al., 2015) and resulted in the values presented in table B1. Other hyperparameters (dropout values and number of neurons in the fully connected layer) also required tuning.

The PIP CNN model (figure A2a) was trained using stochastic gradient descent (SGD) with a batch size of 16 and a decay rate of $10\%$ every five epochs applied to the learning rate. Weights were initialized using the Glorot initialization with a uniform distribution (these are the default settings when using the keras library (Chollet et al., 2015)). The use of a RandomFlip layer (only active during training) as a first layer improved drastically the quality of the training. This layer randomly flips the input image horizontally (left-right flip), vertically (top-bottom flip), both ways or not at all (all four possibilities having the same probability) and thus produces more variety in orientations in the training data. An early stop condition was used in order to end the training, under the condition that the validation loss function did not improve in five epochs. In total 1 634 438 parameters were trained to obtain this model. The performance of the model on the test set is described in figure 4a. Performance is high in general with overall f1-score above $91.1\%$. The worst recognizable class is **HPC** with f1-score of $81.08\%$. The confusion matrix indicates some porosity between **HPC** and **RA**: 8 **RA** identified as **HPC** in the test set ($1.22\%$ of the total). These





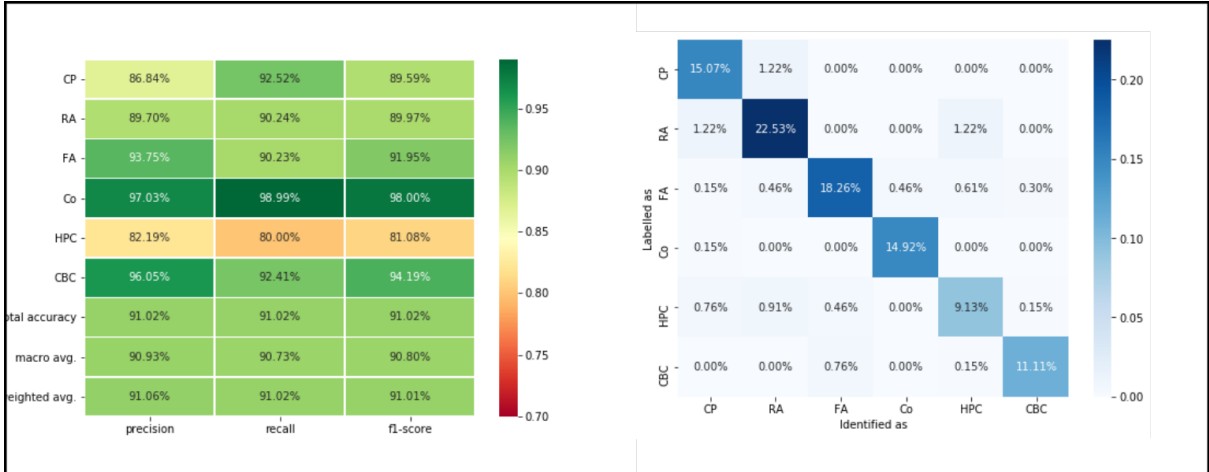

(a) Left, classification report (PIP) obtained on the test set. Right, confusion matrix (PIP) obtained on the test set, values on the diagonal correspond to samples correctly classified. The matrix values are normalized so that they sum up to 100%.

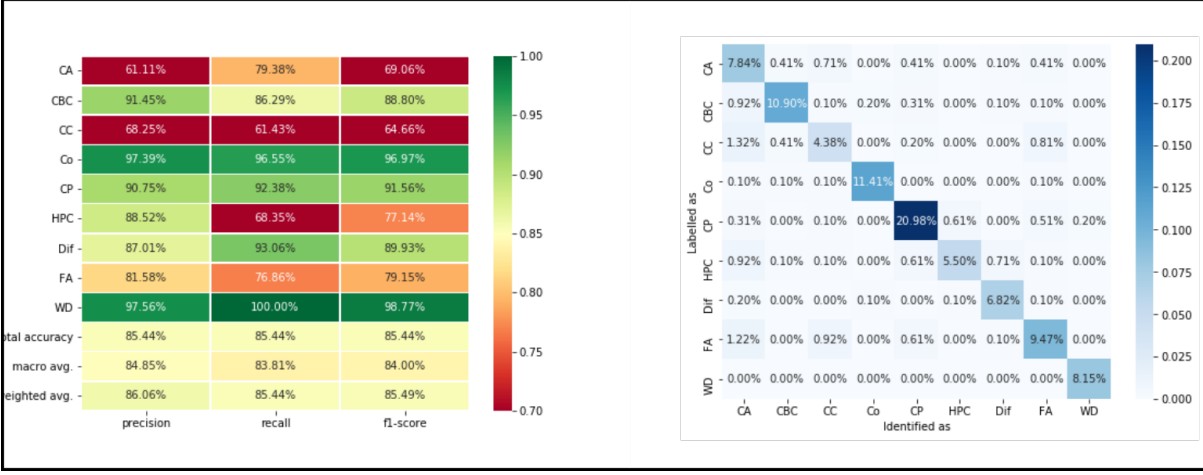

(b) Left, classification report (2DS) obtained on the test set. Right, confusion matrix (2DS) obtained on the test set, values on the diagonal correspond to samples correctly classified. The matrix values are normalized so that they sum up to 100%.

**Figure 4.** Evaluation of training for each probes on an independent test set.

results are hardly comparable with any results found in the literature, since PIP images are not usually used in classification algorithms.

The model corresponding to the 2DS (figure A2b) was trained using the same SGD approach as the PIP model, a batch size of 16 and a decay rate of 10% every five epochs applied to the learning rate. Same weight initialization method as the PIP model was performed and the same RandomFlip data augmentation layer was used during training. Finally, the same early stop condition terminated the training phase. The main difference between the two models was the input size increasing from 110 by 110 for PIP to 200 by 200 for the 2DS (A2b) and the depth of the first convolutionnal layer (64 filters for the 2DS against 32



for the PIP). As a result an additional combination of convolutionnal layer and subsampling layer (and dropout during training) had to be implemented and the size of the fully connected layer of the classifier had to be increased, taking into account that there are now 2048 1 by 1 cells in the final feature maps array at the end of the feature extractor (512 for the PIP). 26 397

129 parameters were determined during the training. The observation of the classification report (left panel of 4b) indicates that some classes are very well identified, which are **CBC**, **Co**, **CP**, **Dif**, and **WD**, while the remaining classes are less well recognized in the test set. Most of the confusion seems to result from images being misclassified in the **CA** class: 18.5% of all **CC** (1.32% of the total), 11.4% of all **HPC** (0.92% of the total) and 9.9% of all **FA** (1.22% of the total) (see right panel of figure 4b). These results exhibit the difficulties we faced to define a set of exhaustive classes with as few overlaps as possible.

When looking at the image examples in table 3, one can easily notice how **CC**, **HPC** and **CBC** classes share similarities in their shapes with the **CA** class, which has much higher internal variability. The most comparable results, we can relate to in the literature are those of Praz et al. (2018). They obtained an overall accuracy of 93.4% for this probe but had two classes less, namely no comparable class to **CC** and one common class merging **CA** and **FA**. If we put together the **CA** and **FA** classes in the confusion matrix, considering that the images confused between the 2 classes are correctly identified (1.22% and 0.41%

of the total), and ignore every image that was either identified or labelled as a capped column (9.15% of the total), the total accuracy reaches 91.1%, which is mainly reflecting how class definitions can affect the results, since the original databases had quite different origins.

## 4    Random inspections: Assessing performance, understanding the results and improving training data

First, the motivations for performing random inspections are given, then the methodology is discussed. Finally, the results are

presented for both probes in the two last subsections.

### 4.1    Motivations and methodology for random inspections

Random inspections have two benefits. The primary benefit is to be able to compare the variability among human predictors and particularly between human predictors and the network.A secondary benefit is simply to produce more manually labelled data. In the case of misclassified images, the newly labelled data can be used to increase intra-class variability and the overall

performance of the network.

In order to compare the implemented CNN algorithm with human performance, ten scientists from the Laboratoire de Météorologie Physique were gathered and given the keys to recognizing the ice crystal classes during two meetings (one for each probe), where they have been presented all morphological classes and given a subset of images from each class from the training data as a reference point. At the end of each meeting they were tested on other images from the training data and did

assist to the correction of their tests. This exercise was thought as a way to improve their skills and as an opportunity to clear some of the confusion that could remain, the results have nonetheless been recorded.





Using data from the recent ICE-GENESIS campaign, 400 images were randomly extracted for the PIP, 500 for the 2DS. An html form was designed and shared with all the participants. They had to attribute a single class associated with a degree of confidence (not taken into account in the scoring) to each image, one after the other. Time spent on each image was recorded.

## 4.2 Results

### 4.2.1 PIP

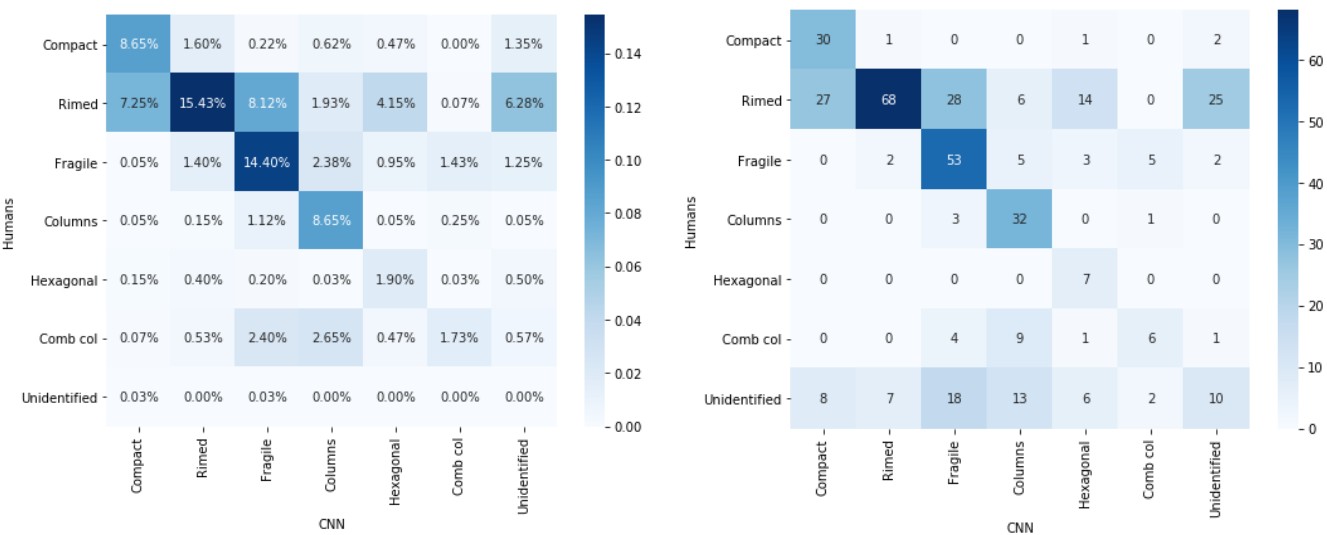

(a) Confusion matrix (PIP), identification threshold at 50% for the CNN results.

(b) Mean confusion matrix (PIP) in numbers, identification threshold at 50% for the CNN and human results

**Figure 5.** Comparison between human and CNN results. Overall, the agreement between them is 50.7%. The expected porosity between **CP/RA** and between **FA/RA** seem to appear and is investigated in Figure A3. Every one of the 40 images considered unidentified by the algorithm show its highest score in the **CP** classe.

On average, each participant spent 1 hour and 10 minutes completing the PIP form. Figure 5 details the overall results of the random inspections, 5a is the confusion matrix normalized so that the values sum up to 100%, 5b present the confusion matrix in numbers. The inspected images belong mostly to **CP, RA, FA** and **Col** classes according to human inspection and CNN.

Humans classified much more particles as **RA** than the algorithm. Most of the images classified as **RA** by humans and not by the CNN are either classified as **CP** or **FA** by the CNN. This confusion was expected since with randomly picked images, the chances were high to find ice crystals in between those classes. When comparing the images, where CNN and humans agree and where they disagree, respectively for the three classes **RA,FA and CP** (Figure A3), it appears that the CNN has developed more consistent class definitions and is therefore superior to the humans in discriminating between the three classes. 25 images

remain unidentified for the CNN and are classified as **RA** by the humans (Figure A4). Looking at their scores, the CNN is





undecided to classify them either in **RA** or **CP** with neither of the two probabilities above 50%. Therefore, one might want to merge **RA** and **CP** before applying the identification threshold in order to have the full estimate of the importance of riming. The porosity between **RA** and **FA** is somewhat less evident with the sampled images. Nevertheless, in order to have a better estimate of the importance of aggregation a similar approach could be applied.

**4.2.2   2DS**

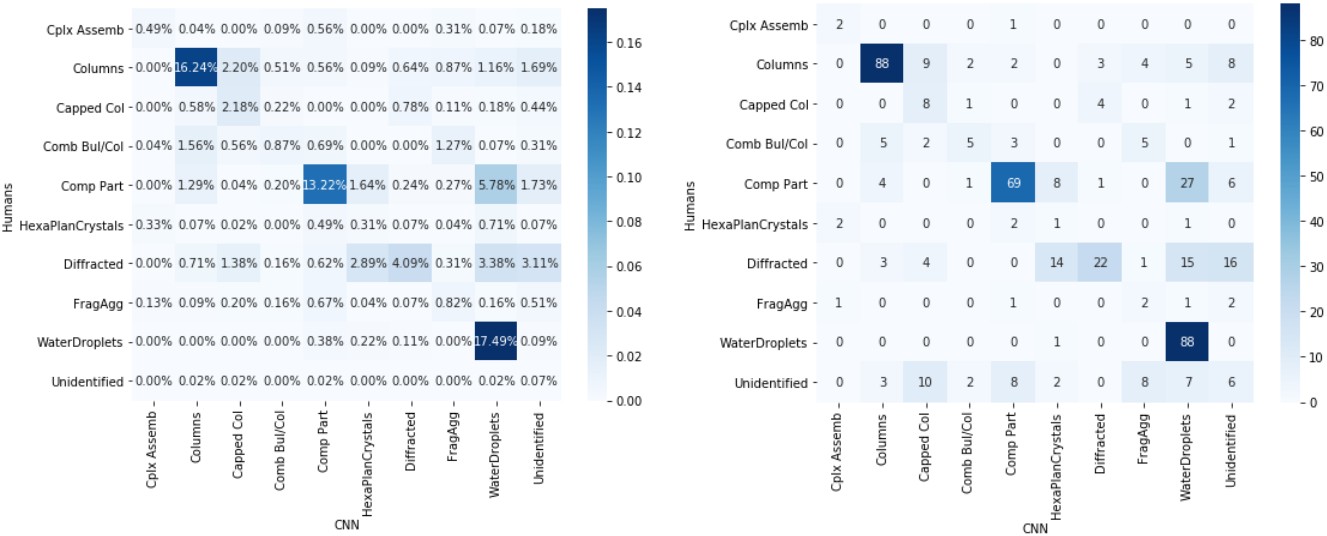

(a) Confusion matrix (2DS), identification threshold at 50% for the CNN results.

(b) Mean confusion matrix (PIP) in numbers, identification threshold at 50% for the CNN and human results

**Figure 6.** Comparison between human and CNN results. Overall, the agreement between them is 58.2%. The expected porosity between **CP/RA** and between **FA/RA** seem to appear and is investigated in Figure A3. Every one of the 40 images considered unidentified by the algorithm show its highest score in the **CP** class.

On average, each participant spent 1 hour and 18 minutes completing the 2DS forms (three forms were provided this time on demand of the participants). Figure 6 details the overall results of the random inspections (same as figure 5 but for the 2DS). The inspected images belong mostly to **WD**, **CP**, **Dif**, and **Col** classes according to both, humans and CNN. With a limited number of classes present in the sample, only four out of nine, a general agreement is found between the CNN and humans
(58.2%). The confusion matrices reveal that the **Dif** class is the most problematic class for the CNN. Indeed, the network spreads **Dif** between **HPC**, **WD** or does not manage to identify them. Additionally, despite being able to identify almost every **WD** as such, the algorithm puts some **CP** in this class in addition to the aforementioned **Dif**. This can be explained by all three classes consisting of possibly small, quasi-spherical or spherical particles. Humans and CNN identified some **CC**. When looking in more details, it seems that both humans and CNN were confused by small, sometimes diffracted sheaths and needles.
Three out of ten participants reported difficulties in classifying 'H' shaped images (shown in figure A5 that we would interpret



as small diffracted columns (see Vaillant de Guélis et al. (2019)). The CNN exhibits this issue as well and shows the lack of such particles in the original training database for the **Dif** class.

## 5 Conclusions

An automatic classification tool has been developed for two OAPs that are routinely used aboard research aircraft in the cloud

observation community in general (Leroy et al., 2017; Defer et al., 2015; Houze et al., 2017; McFarquhar et al., 2011). Both probes, namely the 2DS and the PIP, produce 2D binary images at high frequency in different size ranges. Because of the inability to recognize ice crystal morphology from images with limited number of pixels, the chosen ranges of 300-1280 $\mu$m for the 2D-S and 2000-6400 $\mu$m for the PIP do not overlap. Still they provide us with complementary information and therefore the classification model for both probes is a strong asset for understanding cloud microphysical growth processes.

The methodology presented in this paper was adapted from the most widespread image recognition technique which attempts to reproduce the human's brain ability to learn and recognize shapes: the Convolutionnal Neural Network. Two of these networks have been successfully trained for the two probes and were confronted with inspections by humans on unknown image data. The present study utilized image data from HAIC and EXAEDRE projects in tropical and mid latitude convection (with pronounced crystal growth contributions from aggregation and riming), from AFLUX Arctic project to add vapor diffusion

dominated growth images, and precipitating drops gathered within EUREC4A project in the Carribean Sea. By intention, we didn't tune the methodology neither for a particular type of cloud, nor has it been a goal to add contextual information (of dynamic, thermodynamic, microphysics, or presumed morphological information of crystal populations) to the classification. The human inspection, rarely performed in the scope of applied artificial intelligence, provides a credible evaluation of the CNN tool's performance. The main conclusions of this study are the following:

1. Despite the low number of pixels of OAPs and their binary nature, it is possible for CNNs to learn features associated with the classes defined in section 2.

2. PIP CNN algorithm proved to be more reliable than humans for some classes that see a lot of porosity in field data (e.g Rimed Aggregates, Compact Particles and Fragile Aggregates).

3. Data assimilation has been made possible by running random inspections and should be used for both probes, and

especially for the 2DS, to increase the intra-class variabilities of the few represented habits.

4. Random inspections should be part of the classification routine (see Figure A6), since this allows to quantify its performance, better understand its results, and acquire more labelled data improving the representation of individual classes.

In summary, this study describes a new methodology for ice crystal morphological recognition from OAP images and a way of assessing its performance. Indeed, a systematic and consistent classification of OAP data can provide improved quantitative

information on crystal habits by applying the presented methodology. In the near future, this should facilitate improved detailed microphysical studies, for example targeting habit specific mass relationships (e.g. from Leinonen et al. (2021)). Similar



classification tools can easily be developed for other OAP probes, for example the Cloud Imaging Probe CIP, the four level grey-scale CIP, and the High Volume Precipitation Spectrometer HVPS . The CIP (pixel resolutions of 15 $\mu$m and 25 $\mu$m) mainly overlaps with the 2DS size range, while the HVPS (up to 1.92 cm) would extend the maximum hydrometeor size for

the morphological analysis, as compared to the PIP. Last but not least, a common effort could be made in the global atmospheric sciences community in order to gather a common image database for each instrument, thereby agreeing on defined classes, so that we can develop and test universal future classification algorithms.

*Code and data availability.* Training data (labelled raw images), inspection forms and Python codes can be made publicly available upon request to the authors.



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





## Appendix A: Figures

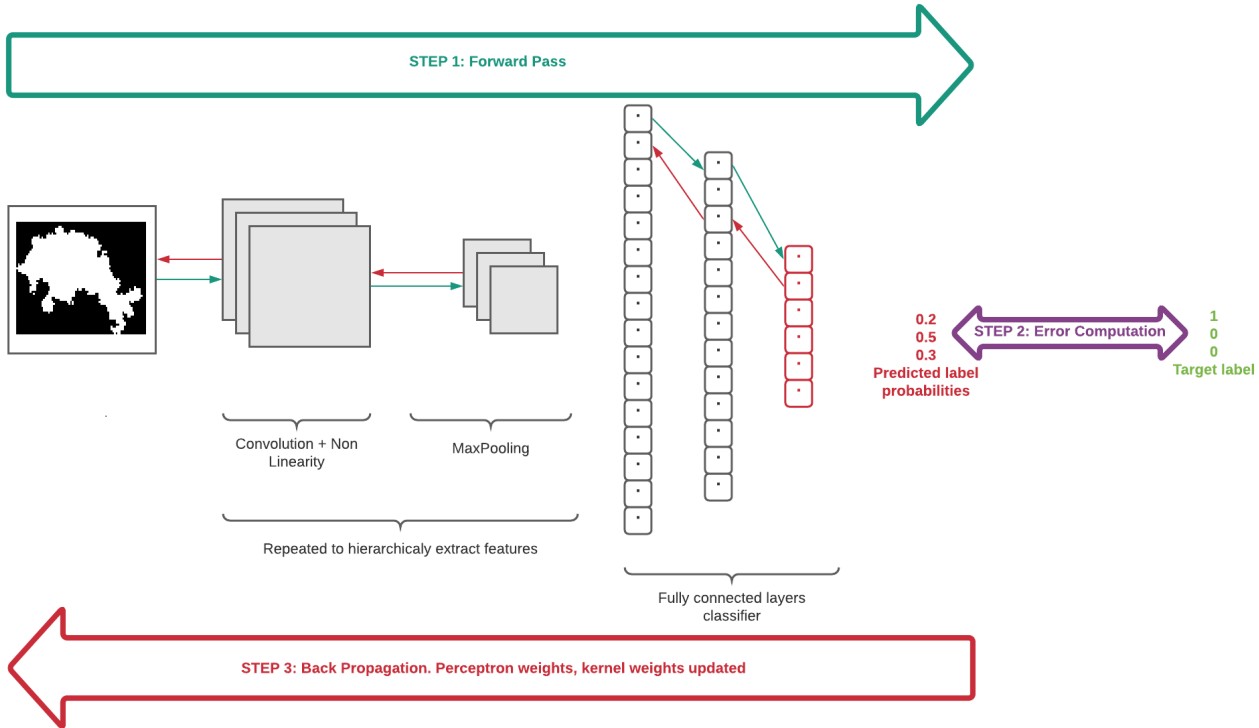

**Figure A1.** The three steps leading to parameter improvement: 1.Forward pass: the image is passed through the network and an output is obtained, 2. An error is computed between this prediction and the target output, 3. This error is propagated by gradient descent back into the network to update the trainable parameters in the model



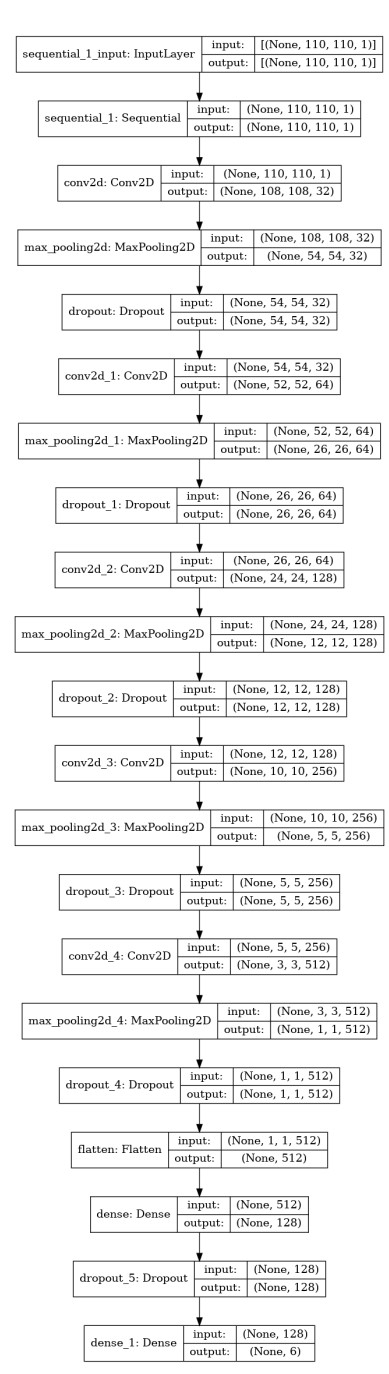

(a) model plot for PIP network

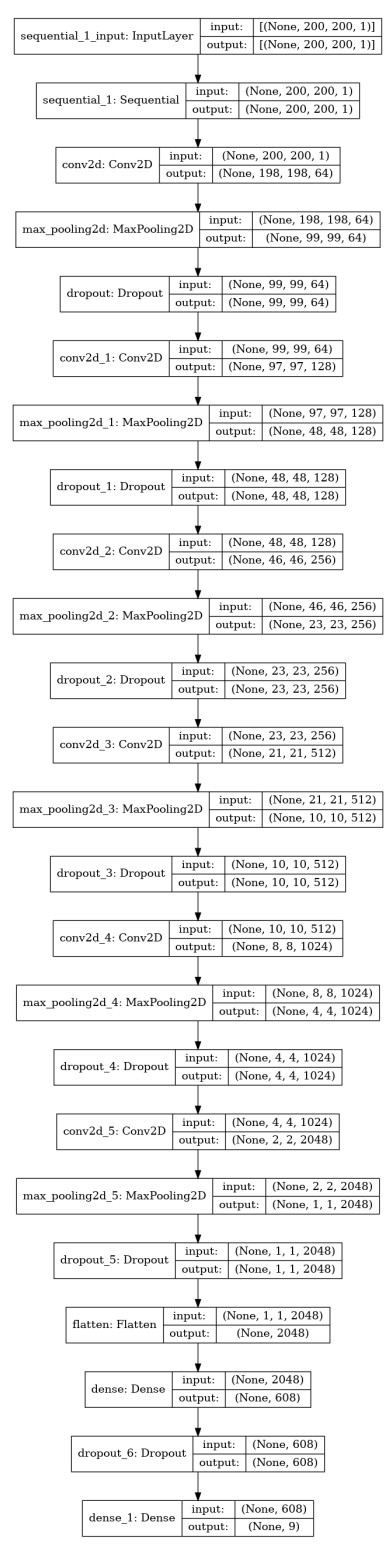

(b) model plot for 2DS network

**Figure A2.** Model plot for each probe





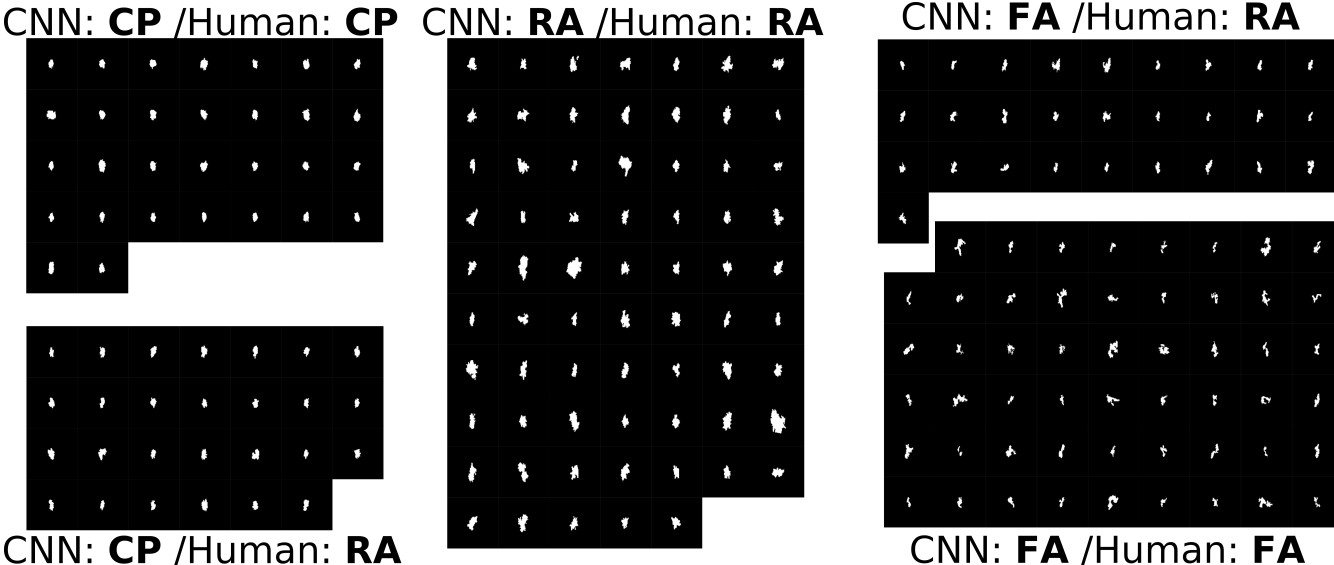

**Figure A3.** Confusion between humans (majority rule) and CNN for the **RA**, **FA** and **CP** classes. The CNN predictions are more consistent than those of humans.



**Figure A4.** Images identified as **RA** by participants, and unidentified by the algorithm. The algorithm gave all these images a high score in both **RA** and **CP**. T





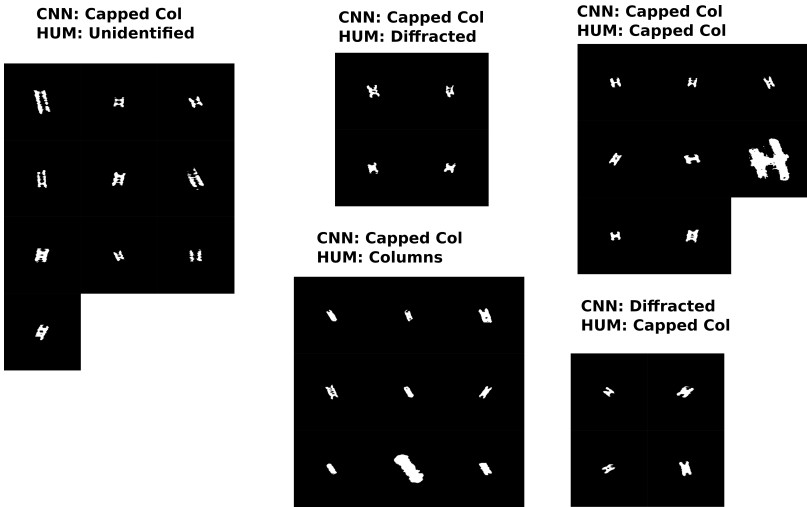

**Figure A5.** Images from the random inspections identified as capped columns by either the CNN or humans.

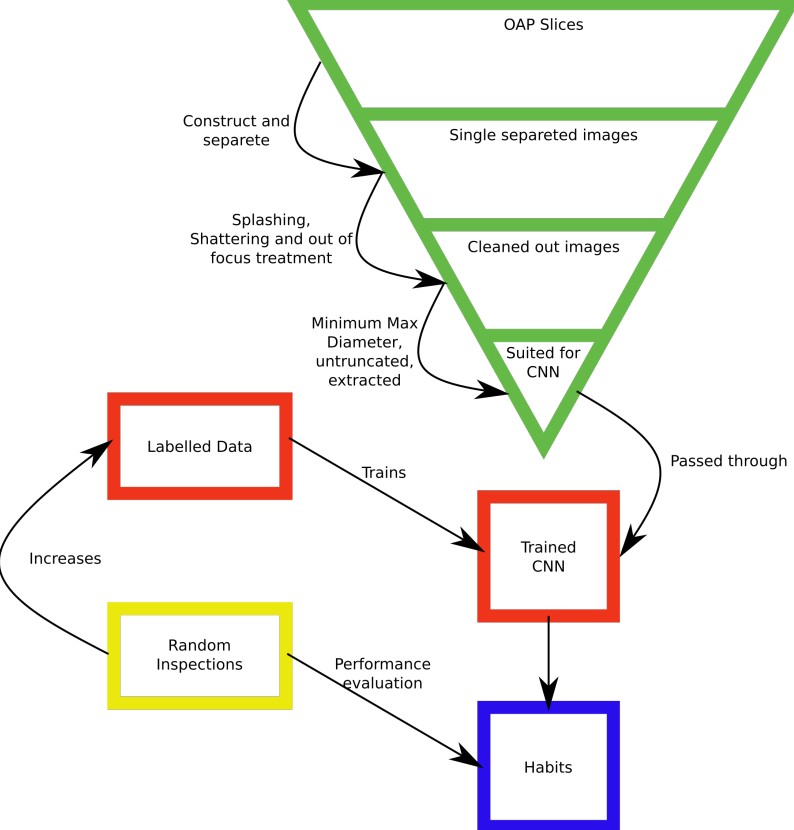

**Figure A6.** Ideal use of the algorithm, which allows for improvements of the training set over time, performance evaluation.





**Appendix B: Table**

**Table B1.** Hyperparameter tuning results for each probe

| Hyperparameter / Layer | Value for PIP model | Value for 2DS model |
|---|---|---|
| dropout | 0.1 | 0.3 |
| dropout_1 | 0.1 | 0.25 |
| dropout_2 | 0.3 | 0.25 |
| dropout_3 | 0.4 | 0.15 |
| dropout_4 | 0.15 | 0.05 |
| dropout_5 | 0.3 | 0.5 |
| dropout_6 | | 0.25 |
| Number of unit in Dense Layer | 128 | 768 |
| Learning rate | 8.031E-4 | 5.055E-4 |



*Author contributions.* Pierre Coutris, Christophe Duroure and Alfons Schwarzenböck formulated the project. Louis Jaffeux developed the methodology and wrote the code to implement and perfect the method. Louis Jaffeux organised the random inspections. Louis Jaffeux wrote the article with contributions from Pierre Coutris, Alfons Scwarzenböck and Christophe Duroure.

*Competing interests.* No competing interests

*Acknowledgements.* The authors would like to thank Angelica Bianco, Céline Planche, Clément Bazantay, Frédéric Tridon, Jean-Luc Barray, Nadège Montoux, Olivier Jourdan, Laurence Niquet for participating to the random inspections of hydrometeor images. Airborne data (from different national and international projects) used in this study were obtained using the aircraft managed by Safire, the French facility for airborne research, an infrastructure of the French National Center for Scientific Research (CNRS), Météo-France and the French National Center for Space Studies (CNES). The microphysical in situ data were collected using instruments from the French Airborne Measurement Platform, a facility partially funded by CNRS/INSU and CNES. This ATR-42 campaign within EUREC4A was funded by the European Research Council (ERC) under the European Union's Horizon 2020 research and innovation programme (EUREC4A Advanced grant No 694768). The French ANR project EXAEDRE was funded by the French Research Ministry (contract ANR-16-CE04-0005). This ICE GEN-ESIS project has received support from the European Union's Horizon 2020 research and innovation programme under grant agreement No 824310 (ICE GENESIS project). Concerning the HAIC-HIWC project, major European campaign and research funding was provided from the European Commission Seventh Framework Program in research, technological development and demonstration under grant agreement ACP2-GA-2012-314314 and the European Safety Agency (EASA) Research Program under service contract EASA.2013.FC27. Major North American funding for flight campaigns was provided by the FAA William Hughes Technical Center and Aviation Weather Research Program, the NASA Aeronautics Research Mission Directorate Aviation Safety Program, the Boeing Co., Environment and Climate Change Canada, the National Research Council of Canada, and Transport Canada. Further funding was provided by the Ice Crystal Consortium. Finally, AFLUX is a joint project of different German universities and research institutes and the LaMP institute of Université Clermont Auvergne (UCA) and the French Research organisation CNRS. AFLUX is embedded in the Transregional Collaborative Research Centre TR 172 (ArctiC Amplification: Climate Relevant Atmospheric and SurfaCe Processes, and Feedback Mechanisms (AC)[3]). The French contribution received financial support from IPEV and CNES.