# Peer review of "Ice crystals images from Optical Array Probes: classification with Convolutional Neural Networks"

_Atmospheric Measurement Techniques, 2022_

## Author Response (AR2)

Atmos. Meas. Tech. Discuss., referee comment RC1 https://doi.org/10.5194/amt-2022-72-RC1, 2022 © Author(s) 2022. This work is distributed under the Creative Commons Attribution 4.0 License.

**Comment on amt-2022-72**

Anonymous Referee #1

Referee comment on "Ice crystals images from Optical Array Probes: classification with Convolutional Neural Networks" by Louis Jaffeux et al., Atmos. Meas. Tech. Discuss., https://doi.org/10.5194/amt-2022-72-RC1, 2022

https://amt.copernicus.org/preprints/amt-2022-72/

**Review:**

Ice crystals images from Optical Array Probes: classification with Convolutional Neural Networks

**By Authors:**

Louis Jaffeux, Alfons Schwarzenböck, Pierre Coutris, and Christophe Duroure

**General comments:**

This study presents a methodology for automatic ice crystal recognition using a Convolutional Neural Network (CNN) machine learning (ML) approach. Two different sensors i.e., the Precipitation Imaging Probe (PIP) and the 2DS-Stereo Probe (2DS), commonly used probes that differ in pixel resolution and measurable maximum size range for hydrometeors collect (binary) black and white images. Ice crystal images from four aircraft in-situ campaigns (EXAEDRE, HAIC, AFLUX, and EUREC4A) were collected to train and evaluate the ML model. The ML method and validation is presented in detail. The results show that the CNN approach show good performance on the test set (mean F1-score > 0.9). Additionally, random inspections were made to compare the variability among human predictors and the CNNs.

Overall, this manuscript is well-written and relevant studies are properly cited. I like the Introduction section and the overview tables, since it is quite comprehensive and reads very friendly for researchers who are not very familiar with this topic. Some parts in results section need further clarification. The scientific benefit of this work, which is to use the method for future microphysical studies, seems reasonable to me. After some minor revisions, I would recommend being published on AMT.

**Major comments:**

The tables on pages 12 and 13 puzzle me a bit. You say (a) is the normalization of (b). Let A be the confusion matrix in numbers. Then the confusion matrix in percentages would be A/sum(A), which does not equal your percentage values. Is there some special way you normalize the confusion matrix values? If so, please explain this step.

Minor comments:

 For classification problems, it is always complicated and labor-intensive to generate large label datasets, with equal amounts of examples for each class. What you could do is to change to a multi-label classification approach, which would reduce your classes and maybe even out the core classes (CP, FA, HPC, RA, Co, WD).

• All figure and table references should start with capital "F" and "T" according to AMT.

■ There are multiple occurrences of "convolutionnal" ï□ change to "convolutional"

 More typos and minor suggestions can be found in the PDF document (marked in red or green or blue). Please also note the supplement to this comment: https://amt.copernicus.org/preprints/amt-2022-72/amt-2022-72-RC1-supplement.pdf

Atmos. Meas. Tech. Discuss., author comment AC1 https://doi.org/10.5194/amt-2022-72-AC1, 2022 © Author(s) 2022. This work is distributed under the Creative Commons Attribution 4.0 License.

Reply on RC1

Louis Jaffeux et al.

Author comment on "Ice crystals images from Optical Array Probes: classification with Convolutional Neural Networks" by Louis Jaffeux et al., Atmos. Meas. Tech. Discuss., https://doi.org/10.5194/amt-2022-72-AC1, 2022

Thank you for your review on our article "Ice crystals images from Optical Array Probes: classification with Convolutional Neural Networks". Adressing your major comment on Figure 5 and 6, I added a clearer description of the mentionned figures in the text below it (sections 4.2.1 and 4.2.2):

"Figure 5 details the overall results of the random inspections, 5a displays all 4000 responses from the ten operators, normalized, while 5b shows how the 400 images are classified by humans and the network in numbers, in this second case a majority rule is used to determine the class attributed by humans, if majority (more than 50%) is not reached for a given image then it is considered unidentified by humans."

The train of thought behind these two plots is to be able to strictly confront human inspections with the network with the first subfigure and to be able to look more closely at images in particular with the second one, such as the ones displayed in the Appendix (A3,A4,A5), in order to see whether the images themselves are problematic, or if the network is producing inaccurate results or even if there were flaws in the way operators were trained.

Concerning your first minor comment on the multi-label classification approach:

you bring up an interesting alternative approach that surely has a couple of advantanges concerning the assumed porosities between our classes.

Finally, thank you for highlighting the typos in the joined PDF, it will make our work that much faster.

Atmos. Meas. Tech. Discuss., referee comment RC2 https://doi.org/10.5194/amt-2022-72-RC2, 2022 © Author(s) 2022. This work is distributed under the Creative Commons Attribution 4.0 License.

**Comment on amt-2022-72**

Anonymous Referee #2

Referee comment on "Ice crystals images from Optical Array Probes: classification with Convolutional Neural Networks" by Louis Jaffeux et al., Atmos. Meas. Tech. Discuss., https://doi.org/10.5194/amt-2022-72-RC2, 2022

Review of AMT\_2022\_72

The manuscript is nicely organized and written (minus some typos/spelling that do not detract from the narrative). The description of the CNN is very clearly laid out and easy to follow (even for a non-expert in ML). The approach to doing fewer categories is smart and I think yields more robust results that will be useful for other datasets. I really like Table 3 – it is an excellent visual to help understand the classification method.

In general – I would say the manuscript needs some minor revisions before publication. I would recommend a careful re-read of the revised manuscript for spelling, typos, and clarity.

Major Comments:

- It is mentioned that ~8000 images are used from 4 different field campaigns. However, it is not discussed how many individual precipitating events are included (1 from each campaign?). I am curious about how much biasing or co-variability could be potentially introduced when training the dataset and knowing how many separate events were used would help better understand the breadth of environmental conditions.
- Related there is not much description about the airborne campaigns from which the data originates (there is a brief mention in the conclusion). It would be helpful to have some more context earlier on in the paper.
- I missed where it is detailed how many particles are represented in each category from the datasets – for the training, do you have similar numbers of particles for each class type represented?

The matrices in Fig. 5 and 6 are a little hard to understand. Could you please add some details to the description or clarify as to how you are calculating these percentages?

Minor Comments:

- There are minor mis-spellings throughout the document (e.g., convolutionnal, functionnality, etc.). There are some passive phrases and word usages that could use revisiting. An additional readthrough by the co-authors should be able to address these issues.
- Page12, L242 245: This sentence is unclear and very long. Could you split it up and clarify the points here?

Atmos. Meas. Tech. Discuss., author comment AC2 https://doi.org/10.5194/amt-2022-72-AC2, 2022 © Author(s) 2022. This work is distributed under the Creative Commons Attribution 4.0 License.

**Reply on RC2**

Louis Jaffeux et al.

Author comment on "Ice crystals images from Optical Array Probes: classification with Convolutional Neural Networks" by Louis Jaffeux et al., Atmos. Meas. Tech. Discuss., https://doi.org/10.5194/amt-2022-72-AC2, 2022

Thank you for your review on our article "Ice crystals images from Optical Array Probes: classification with Convolutional Neural Networks".

First and second major comments: Few lines added in the caption of Table 2 (could also be added to third point of Section 2)

"All the PIP images in the original data set originate from 2 events in the EXAEDRE campaign and 3 events of the HAIC campaign. The context of these events are thunderstorms in the Mediteranean Sea for EXAEDRE and Mesoscale Convective Systems in french Guiana for HAIC. Most of the 2DS data also originate from the same events. The AFFLUX campaign data was extracted from a single flight in Arctic clouds to provide more Col, CBC and HPC of various sizes. Finally, all the water droplets of the 2DS data set were captured during a single flight in liquid water clouds in the Carribean Sea during the EUREC4A project."

Third major comment: table 3 shows the number of images in each category in parenthesis

Fourth major comment on Figure 5 and 6: I added a clearer description of the mentionned figures in the text below it (sections 4.2.1 and 4.2.2):

"Figure 5 details the overall results of the random inspections, 5a displays all 4000 responses from the ten operators, normalized, while 5b shows how the 400 images are classified by humans and the network in numbers, in this second case a majority rule is used to determine the class attributed by humans, if majority (more than 50%) is not reached for a given image then it is considered unidentified by humans."

The train of thought behind these two plots is to be able to strictly confront human inspections with the network with the first subfigure and to be able to look more closely at images in particular with the second one, such as the ones displayed in the Appendix (A3,A4,A5), in order to see whether the images themselves are problematic, or if the network is producing inaccurate results or even if there were flaws in the way operators were trained.

Second minor comment: we suggest an alternative text

"Appendix Figure A3 shows the crystal images that are referred to in Figure 5b, for CP,RA and FA classes. Inspecting these sets of images, **CNN: CP/ Human: RA** are objectively closer to **CNN: CP/ Human: CP** than to **CNN: RA/ Human: RA**, and **CNN: FA/ Human: RA** is arguably closer to **CNN: FA/ Human: FA** than to **CNN: RA/ Human: RA**. It appears that the CNN has developed more consistent class definitions and is therefore superior to the humans in discriminating between the three classes."

Using the Figure A3 so directly might also require that we move this Figure from the Appendix to the main text.